# The Impact of Nutritional Support on Outcomes of Lung Cancer Surgery—Narrative Review

**DOI:** 10.3390/jcm14093197

**Published:** 2025-05-05

**Authors:** Alicja Werblińska, Dominika Zielińska, Lidia Szlanga, Piotr Skrzypczak, Maciej Bryl, Cezary Piwkowski, Piotr Gabryel

**Affiliations:** Department of Thoracic Surgery, Poznan University of Medical Sciences, Szamarzewskiego 62 Street, 60-569 Poznan, Polandpiotr.j.skrzypczak@gmail.com (P.S.); mbryl@wcpit.org (M.B.);

**Keywords:** lung cancer, malnutrition, nutritional assessment, surgery, ERAS protocol

## Abstract

**Background:** Malnutrition is a prevalent yet often overlooked issue in lung cancer patients, significantly affecting surgical outcomes. This review examines the impact of nutritional status on lung cancer surgery and explores the role of nutritional assessment and intervention strategies. **Methods:** A comprehensive literature search was conducted using databases such as PubMed, Scopus, and Web of Science. Key studies on nutritional status assessment, preoperative nutritional support, and their impact on surgical outcomes were analyzed. **Results:** Malnutrition in lung cancer patients is associated with increased postoperative complications, prolonged hospital stays, and reduced survival rates. Various assessment tools, including dietary interviews, physical examinations, laboratory tests, and body composition analyses, can help identify malnourished patients. Nutritional support strategies such as high-protein diets, oral supplements, enteral and parenteral nutrition, and perioperative immunomodulation improve clinical outcomes. **Conclusions:** Implementing standardized nutritional assessment and support protocols is crucial for optimizing surgical outcomes in lung cancer patients. Integrating these strategies into the Enhanced Recovery After Surgery (ERAS) protocol may further enhance recovery and long-term prognosis.

## 1. Introduction

Lung cancer is one of the most common causes of cancer-related morbidity and mortality worldwide [1,2]. Apart from the histological type, the most important factor determining the treatment method and prognosis is cancer stage [3]. It has also been shown that other factors, such as age, gender, and comorbidities, can influence the qualification for specific therapeutic approaches and determine both early and long-term outcomes of treatment [4,5]. In recent years, there has been increasing attention to the role of nutritional status assessment and nutritional interventions in cancer patients. Epidemiological data indicate that malnutrition may affect up to 85% of patients with advanced cancer [6]. Lung cancer ranks fourth, following pancreatic, head and neck, and esophageal cancers, in terms of the risk of malnutrition occurrence [7]. It is estimated that up to 70% of lung cancer patients may experience weight loss, particularly unfavorable loss of muscle mass [8]. At the same time, nutritional status assessment and nutritional interventions are undertaken too infrequently [9]. Malnutrition in lung cancer patients is therefore a significant issue yet is often overlooked in daily clinical practice, which may potentially affect treatment outcomes.

Currently, the most effective treatment for lung cancer is surgery. The best results, especially in the early stages of the disease, are achieved through minimally invasive complete anatomical lung resection with lymphadenectomy. With the introduction of a low-dose computed tomography screening program for high-risk groups, the number of patients diagnosed in early stages has increased, and long-term treatment outcomes have also improved [10]. Due to the early stage of the disease, the risk of malnutrition in the group of surgically treated patients is lower than in the entire lung cancer patient group, but it may still influence early and long-term outcomes [11]. A breakthrough in oncology treatment, currently occurring through the introduction of targeted molecular therapy and immune checkpoint inhibitors, has enabled effective oncological surgery even in patients with more advanced stages of cancer [12]. Given the higher risk of malnutrition and the significant burden on the body in combined treatments, nutritional status assessment and appropriate interventions may be particularly important in this group. Proper nutritional preparation of patients can improve their nutritional status, reduce postoperative complications, and shorten hospital stay [13,14,15,16].

Despite research indicating the importance of preoperative nutritional assessment and appropriate nutritional preparation, there are still many knowledge gaps in this area. These include the most effective methods of assessing nutritional status, optimal nutritional strategies, and the importance of immunomodulatory nutrition. In the context of lung cancer surgery, particularly significant yet poorly studied are issues related to nutrition in patients with early-stage lung cancer undergoing perioperative immunosuppressive treatment as well as the role of nutrition as part of the ERAS protocol.

The aim of this paper was to assess the impact of nutritional support on surgical treatment outcomes in lung cancer patients. This review aimed to evaluate the role of nutritional assessment and support in optimizing short- and long-term outcomes in patients undergoing surgery for lung cancer.

## 2. Nutritional Status Assessment in Patients with Lung Cancer

### 2.1. Nutritional Status of Cancer Patients

Malnutrition is defined as a loss of 10% of body weight or a low body mass index (BMI < 18). This definition may slightly differ depending on the guidelines (e.g., ASPEN, ESPEN, etc.). However, modern diagnostic approaches such as the Global Leadership Initiative on Malnutrition (GLIM) provide more comprehensive criteria, requiring both phenotypic and etiological indicators for diagnosis [17].

According to GLIM, malnutrition is confirmed when at least one phenotypic criterion (e.g., weight loss > 5% within 6 months, low BMI, or reduced muscle mass) and one etiological criterion (e.g., reduced food intake or inflammation due to disease) are met [17]. 

The causes of malnutrition are diverse, but in cancer patients, malnutrition often results from insufficient dietary intake due to anorexia (loss of appetite), nausea, taste alterations, eating difficulties (e.g., dysphagia or rapid fatigue), gastrointestinal disturbances (e.g., mucosal inflammation or diarrhea), increased energy expenditure, and psychological factors (e.g., depression or anxiety) [6]. Additionally, it is crucial to identify patients with cancer cachexia in a timely manner. Cancer cachexia is a complex condition characterized by weight loss due to reduction in both muscle mass and fat tissue. Cachexia is associated with chronic inflammation and metabolic disturbances, which influence its pathogenesis. It affects up to 39% of patients with non-small-cell lung cancer and significantly increases mortality risk [18,19].

### 2.2. Methods of Nutritional Status Assessment

Assessing the nutritional status of lung cancer patients is crucial for implementing appropriate nutritional interventions and improving treatment outcomes. Several tools and methods can be used for this purpose [20,21,22]. The choice of method depends on the individual needs of the patient and clinical specifics [23]. A dietary interview is easy to conduct and cost-effective, but it relies on the patient’s memory and their ability to report accurately [23]. Physical examination allows for the quick detection of signs of malnutrition, but its effectiveness depends on the examiner’s experience [24]. Laboratory tests provide objective data; however, their results may be influenced by inflammatory conditions and comorbidities [23]. Scales and questionnaires are quick and non-invasive but may not address all aspects of nutritional status [25]. Body composition analysis provides precise data on muscle and fat mass but requires specialized equipment [24]. The CONUT and CALLY indices are based on laboratory tests and can support the assessment of patient prognosis but do not include anthropometric parameters. Each method has its limitations, so a multifaceted approach is recommended [26,27]. The integration of various methods allows for a more comprehensive evaluation of the patient and more effective implementation of nutritional interventions. Combining a dietary interview, physical examination, laboratory tests, and body composition assessment provides the most reliable picture of the patient’s nutritional status [23,24]. Table 1 presents an overview of advantages and disadvantages of these various nutritional assessment methods.
Dietary interview: The dietary interview is the basic tool for assessing nutritional status. It includes a detailed conversation with the patient about eating habits, such as the frequency and types of meals consumed, dietary preferences, and the use of dietary supplements [28]. Additionally, the interview includes questions about eating difficulties, such as swallowing problems (dysphagia), nausea, vomiting, diarrhea, constipation, and changes in taste and smell perception [28,29]. An important element is also the assessment of body weight changes, especially losses exceeding 10% of initial body weight within the last 3–6 months [22]. The dietary interview allows for the initial identification of patients at risk of malnutrition and the adaptation of nutritional interventions to their individual needs [20].Physical examination: Physical examination is essential for assessing the overall condition of a patient undergoing surgery for lung cancer and detecting signs of malnutrition or cancer cachexia [30,31,32]. It includes the following:
Measurement of body weight and height: calculation of body mass index (BMI), where BMI < 18.5 kg/m^2^ indicates malnutrition [33,34];Measurement of calf and arm circumference: calf circumference <33 cm in women and <34 cm in men may indicate malnutrition [30,31], and arm circumference <22 cm in women and <23.5 cm in men is also associated with poorer nutritional status [35];Assessment of muscle strength: handgrip strength test is a simple tool for evaluating muscle strength, which is an important indicator of nutritional status, and values below 16 kg in women and 27 kg in men may suggest sarcopenia [30,32];Clinical signs of malnutrition: dry skin, brittle hair, edema, changes in mucous membranes, and muscle weakness [6].
Laboratory tests: laboratory tests provide objective data regarding nutritional status; however, their interpretations necessitates consideration of inflammatory conditions and comorbidities, which can influence the results [36]. Key parameters include the following:
Albumin (half-life 20 days): low albumin levels (<3.5 g/dL) correlate with poorer prognosis but exhibit low sensitivity to short-term nutritional changes due to its long half-life [37];Prealbumin: with a short half-life (2–3 days), prealbumin is more sensitive marker of current nutritional status; however, its deficiency can also result from the acute phase response, and levels below 10 mg/dL suggest malnutrition [38];Transferrin: low transferrin levels (<200 mg/dL) may indicate iron and protein deficiency, but its concentration is dependent on inflammatory status [39];Peripheral lymphocytes: reduced lymphocyte count (<1500/μL) is associated with impaired immune function but may be associated with chemotherapy or infections [40];Cholesterol: low cholesterol levels (<97 mg/dL) increase mortality risk in patients [41];CRP (C-reactive protein): levels above 10 mg/L indicate inflammation, often associated with cancer cachexia [42].Laboratory tests are useful in monitoring the response to nutritional interventions, but they should be interpreted in conjunction with clinical assessment (e.g., nutritional history, anthropometric measurements, etc.) and screening tools (e.g., NRS 2002, PG-SGA, etc.) to avoid misdiagnosis.
Scales and questionnaires: scales and questionnaires are simple tools for assessing nutritional status and the risk of malnutrition [21]. Some of the most commonly used include the following:
Mini Nutritional Assessment (MNA): A tool primarily for older adults, assessing nutritional status based on interview, physical examination, and laboratory parameters. A score < 17 indicates malnutrition, 17–23.5 indicates risk of malnutrition [43];Subjective Global Assessment (SGA): a scale based on clinical interview and physical examination, used to assess nutritional status and malnutrition risk [44,45];Patient-Generated Subjective Global Assessment (PG-SGA): a modified version of the SGA that includes the patient’s assessment of their nutritional status and cancer-related symptoms [46,47];NRS 2002 (Nutritional Risk Screening 2002): A screening tool for assessing malnutrition risk in hospitalized patients, covering weight loss, food intake, and disease severity. A score ≥ 3 indicates malnutrition risk [48];MUST (Malnutrition Universal Screening Tool): A simple tool for assessing malnutrition risk, applicable to various patient groups, including oncology patients. A score ≥ 2 indicates high malnutrition risk [49];Global Leadership Initiative on Malnutrition (GLIM): This diagnostic framework utilizes a two-step approach for identifying malnutrition. The process begins with initial screening using validated tools such as NRS 2002 or MUST. For patients identified as at-risk, a comprehensive assessment evaluates both phenotypic and etiologic criteria.For diagnosis, patients must meet at least one phenotypic criterion: unintended weight loss exceeding 5% within six months or 10% over a longer period, low body mass index (BMI below 18.5 for patients under 70 years or below 20 for those 70 and older), or clinically evident reduction in muscle mass. Additionally, at least one etiologic criterion must be present: either significantly reduced food intake (consuming less than 50% of nutritional requirements for more than one week) or presence of inflammation as evidenced by markers like elevated CRP in cancer patients [17].The GLIM framework not only facilitates malnutrition diagnosis but also enables severity stratification (Stage 1 or 2) and has demonstrated particular utility in oncology populations through validation studies. This standardized approach enhances consistency in nutritional assessment across clinical settings while addressing the complex interplay between nutritional status and disease processes [17].
Body composition analysis: body composition analysis provides detailed information about muscle and fat mass, which is important in detecting malnutrition or cancer cachexia [20,30]. Common methods include the following:
Bioelectrical impedance analysis (BIA) represents a non-invasive and readily accessible body composition assessment modality that is gaining increasing clinical relevance in the nutritional monitoring of oncological patients. This technology enables comprehensive quantitative and qualitative evaluation of muscle mass through measurement of fat-free mass (FFM) and skeletal muscle mass (SMM), with normative values corresponding to 75–85% of FFM in healthy adult populations. A particularly significant parameter is the phase angle (PA), which serves as a biomarker of cellular membrane integrity. Reference values for PA typically range between 5–7° in healthy individuals, while diminished values (<4.5°) demonstrate significant correlation with poorer prognostic outcomes in neoplastic disease [50]. In pulmonary carcinoma specifically, phase angle measurements below 3.8° are associated with a 2.5-fold increase in mortality risk, establishing this parameter as a valuable prognostic indicator. BIA facilitates early detection of muscle mass depletion (>5% reduction in FFM over a 3-month period) and enables identification of sarcopenia, including in patients with normal or elevated body mass index (BMI) [50]. The skeletal muscle mass index (SMI) proves particularly clinically relevant, with values below 7.26 kg/m^2^ in male patients and 5.45 kg/m^2^ in female patients correlating with a threefold increased risk of postoperative pulmonary complications. This methodology also proves instrumental in monitoring therapeutic outcomes of nutritional interventions and prehabilitation programs through objective assessment of muscle mass accrual. Beyond muscular evaluation, BIA provides critical data regarding adipose tissue content (reference ranges: 10–20% in males; 20–30% in females) [51];Dual-energy X-ray absorptiometry (DXA) currently represents the gold standard for body composition assessment, utilizing the differential absorption of X-rays at two distinct energy levels. This technology enables simultaneous and precise evaluation of fat-free mass with 1–2% accuracy, adipose tissue with differentiation between subcutaneous and visceral depots, and bone mineral content [52]. In oncological practice, DXA has proven particularly valuable for early detection of cancer cachexia, where a >5% loss of muscle mass over 3 months is considered clinically significant, and for diagnosing sarcopenia using established thresholds of FFM <17 kg/m^2^ in men and <15 kg/m^2^ in women [52]. Characterized by low radiation exposure (1–10 μSv) and rapid scan times (10–15 min), this method also permits regional fat distribution analysis, which is crucial for monitoring body composition changes during therapy. However, results may be influenced by hydration status, and limitations exist in assessing individuals with body weight exceeding 130 kg [53];CT and MRI: For cases requiring more precise evaluation, particularly in surgical planning, computed tomography (CT) and magnetic resonance imaging (MRI) techniques are employed. CT, especially when analyzing a single slice at the third lumbar vertebra (L3), is considered the reference standard for quantitative muscle mass assessment, offering measurement error below 2% while simultaneously evaluating muscle quality through Hounsfield unit (HU) analysis. In lung cancer, established skeletal muscle index (SMI) thresholds of <52.4 cm^2^/m^2^ for men and <38.5 cm^2^/m^2^ for women are particularly relevant, along with myosteatosis criteria defined as <41 HU for men and <38 HU for women. Research demonstrates that >8.3% muscle mass loss on CT correlates with a 4.1-fold increased risk of postoperative complications, which is critical for thoracic surgery patient selection [54].MRI, as a non-ionizing radiation modality, offers even broader diagnostic capabilities, including whole-body muscle volumetry with ±1.5% accuracy and quantitative fat infiltration assessment (PDFF) with ±0.5% precision. Advanced MRI protocols incorporate diffusion tensor imaging for muscle microstructure evaluation and spectroscopy for metabolic assessment through ATP/PCr ratio measurement. Clinically, the detection of even early muscular changes is particularly significant, where fat infiltration exceeding 5% on PDFF correlates with substantially worse prognosis. Importantly, results obtained through different modalities show strong correlations—for instance, reduced phase angle values in BIA (<4.5°) correlate with decreased SMI on CT (r = 0.82), while increased BIA resistance corresponds to greater fat infiltration visible on MRI [54]. In clinical practice, a stepped approach is therefore recommended, with BIA serving as a screening tool, while DXA, CT, or MRI are utilized for more detailed assessment based on specific diagnostic needs and technical availability. This integrated methodology optimizes the unique advantages of each technique while minimizing their respective limitations [55].
CONUT (Controlling Nutritional Status): The CONUT index is a simple tool based on laboratory parameters such as albumin levels, peripheral lymphocyte count, and cholesterol levels. A high CONUT score (≥6) is associated with worse prognosis, including lower overall survival (OS) and disease-free survival (DFS). A low CONUT score may indicate better prognosis [26,56].CALLY (C-reactive protein-albumin-lymphocyte): The CALLY index is specifically designed for assessing nutritional status in cancer patients. A score <3.0 is associated with worse DFS and OS. The index includes the assessment of inflammatory markers—C-reactive protein, albumin, and lymphocytes. The CALLY index helps in the early detection of cancer cachexia and the implementation of appropriate nutritional interventions [27,57].

## 3. The Impact of Malnutrition on Outcomes of Lung Cancer Surgery

Malnutrition is a significant risk factor for poorer surgical outcomes in lung cancer patients [58]. It affects immune system function, wound healing, hospital length of stay, mortality, and the ability of patients to undergo postoperative rehabilitation. Studies show that malnourished patients (BMI < 18.5 or >10% weight loss) have a 2–3 times higher risk of postoperative complications, including lung infections, respiratory failure, and death [59,60]. The average hospital stay is 2–11 days longer compared to well-nourished patients [61]. Prolonged hospitalization is associated with higher treatment costs and a greater burden on the healthcare system [62]. Malnutrition weakens immune function, making surgical lung cancer patients more susceptible to infections. Research indicates that malnourished patients have a significantly higher risk of postoperative infections, such as pneumonia, surgical wound infections, and sepsis [63]. Malnutrition also leads to weakened respiratory muscles, impairing airway clearance and increasing the risk of lung infections [64,65]. Protein and nutrient deficiencies further impair wound healing, increasing the risk of infections [66]. Malnutrition is a strong risk factor for increased mortality in lung cancer patients undergoing surgery [62]. Malnourished patients have a higher risk of death within 30 days postoperatively compared to well-nourished patients [62]. Malnutrition also affects long-term survival, with malnourished patients showing lower overall survival (OS) and disease-free survival rates (DFS) [67]. The increased mortality is attributed to factors such as overall physical weakness, greater susceptibility to infections, impaired regenerative processes (including wound healing), and cancer progression associated with cachexia [68]. Malnutrition, particularly in the context of cancer cachexia, leads to muscle loss and reduced muscle strength [69]. This has significant consequences for post-lung cancer surgery patients [70,71]. Muscle weakness complicates respiratory exercises and coughing, increasing the risk of pulmonary complications such as pneumonia and respiratory failure [11]. Muscle loss negatively impacts patients’ quality of life and their ability to return to normal activities. Weak respiratory muscles and overall physical weakness increase the risk of postoperative complications, such as pulmonary embolism or cardiopulmonary failure [72]. Studies show that patients with cancer cachexia have approximately 30% lower survival rates compared to those without cachexia [73]. Additionally, cachexia may be responsible for at least 20% of deaths among cancer patients [74]. Early implementation of nutritional and pharmacological interventions can slow the progression of cachexia and improve prognosis [75]. As illustrated in Figure 1 Nutritional Support Enhances Surgical Outcomes in Lung Cancer, proper nutritional support can significantly improve recovery and reduce complications. Malnutrition and associated muscle loss significantly impact the quality of life of post-lung cancer surgery patients. Malnourished patients struggle with daily activities such as walking or getting out of bed [76]. Malnutrition can also lead to depression, anxiety, and mood disorders, further worsening prognosis [57,77,78].

## 4. Preoperative Nutritional Strategies

### 4.1. Nutritional Support Strategies for Lung Cancer Patients

Nutritional support is a key component of comprehensive lung cancer treatment, playing a crucial role in improving therapeutic outcomes, reducing complications, and increasing survival rates. Inadequate nutrition can lead to physical weakness, increased infection risk, and prolonged postoperative recovery [57].

Nutritional strategies focus on achieving short- and long-term goals aimed at improving the patient’s nutritional status, reducing complication risks, and enhancing the effectiveness of oncological treatment [79].

#### 4.1.1. Short-Term Goals

Short-term goals include improving preoperative nutritional status by increasing muscle mass and correcting biochemical parameters such as albumin and prealbumin levels, which are key markers of nutritional status [36]. Another important aspect is reducing the risk of postoperative complications, including preventing infections, accelerating wound healing, and reducing the risk of respiratory failure [80]. Optimizing nutritional status also shortens hospital stays, benefiting both the patient and the healthcare system [81].

#### 4.1.2. Long-Term Goals

Long-term goals include improving survival rates, both in terms of overall survival (OS) and disease-free survival (DFS). Additionally, appropriate nutritional support can counteract cancer cachexia, which leads to significant patient weakness and reduced quality of life [75]. Supporting postoperative rehabilitation by increasing muscle strength and overall physical fitness enables faster recovery to daily activities [80,81].

### 4.2. Forms of Nutritional Support

Nutritional support for cancer patients includes various strategies tailored to individual needs and clinical conditions [82].

#### 4.2.1. High-Protein Diet

A high-protein diet is crucial for preparing patients for surgery, especially those at risk of malnutrition. Protein plays a vital role in maintaining muscle mass, which is essential for postoperative recovery. Recent studies suggest an optimal protein intake of 1.2–1.5 g/kg body weight/day, but for severely malnourished patients or those at high risk of muscle loss (e.g., elderly patients), 1.5–2.0 g/kg body weight/day may be recommended [28,83]. Protein sources include lean meats (poultry and beef), fish (especially fatty fish rich in omega-3 fatty acids), eggs, dairy (cheese, yogurt, and milk), and plant-based sources such as legumes (beans, lentils, and chickpeas), tofu, and quinoa [84]. For patients with limited tolerance for solid foods, liquid protein supplements, such as whey protein isolates, may be considered [85]. Recent studies suggest that plant-based proteins can be as effective as animal proteins in supporting muscle recovery, particularly in patients with chronic conditions such as diabetes or heart disease. Additionally, plant-based proteins may benefit the gut microbiome, which is important for overall health [84].

#### 4.2.2. Oral Nutritional Supplements

Oral nutritional supplements are particularly important for patients unable to consume sufficient food due to appetite loss, swallowing difficulties, or other health issues [83]. High-protein oral supplements can prevent an average weight loss of 2 kg [86]. Studies show that protein supplementation combined with resistance exercises (if the patient’s condition allows) can significantly improve muscle mass and strength before surgery, leading to better postoperative outcomes [85,87].

#### 4.2.3. Enteral Nutrition

When oral nutrition is insufficient or impossible, enteral nutrition is used [88]. This method involves nasogastric tubes or percutaneous endoscopic gastrostomy (PEG), which is used for patients with swallowing difficulties or chronic malnutrition [89]. Enteral nutrition in lung cancer patients with oral intake difficulties reduces the risk of infectious complications and improves wound healing. It is recommended to initiate enteral nutrition 7–10 days before surgery in malnourished patients [82]. Advantages of enteral nutrition include better control of energy balance, reduced infection rates related to malnutrition, and lower risk of bacterial translocation in the gastrointestinal tract compared to parenteral nutrition [90]. PEG is preferred for patients requiring long-term nutritional support, as it provides more stable access to the gastrointestinal tract and reduces the risk of aspiration compared to nasogastric tubes [91]. However, enteral nutrition carries risks such as diarrhea, tube obstruction, and aspiration pneumonia, particularly in patients with impaired swallowing reflexes [92]. Additionally, PEG placement carries risks of infection at the insertion site and bleeding, requiring careful patient selection and monitoring [93]. Despite these limitations, enteral nutrition remains the preferred method for patients with functional gastrointestinal tracts, as it better mimics natural digestion and nutrient absorption [94].

#### 4.2.4. Parenteral Nutrition

Parenteral nutrition is used when gastrointestinal function is impaired and enteral nutrition is not feasible [82]. This method delivers all necessary nutrients intravenously, which is crucial for patients with severe malabsorption, intestinal obstruction, or short bowel syndrome [95]. The main advantage of parenteral nutrition is precise control over macronutrient and electrolyte delivery, allowing tailored therapy to individual patient needs [96]. It is also effective for critically ill patients, where malnutrition can worsen prognosis [97]. However, parenteral nutrition carries risks of serious complications, such as sepsis related to vascular access, metabolic disturbances (hyperglycemia and micronutrient deficiencies), and liver damage with long-term use [98]. Additionally, parenteral nutrition is more costly and requires intensive laboratory monitoring to avoid metabolic complications. Therefore, it is recommended only when enteral nutrition is not feasible or insufficient [99,100].

#### 4.2.5. Perioperative Immunomodulation

Immunomodulation is a nutritional strategy aimed at improving immune function and reducing inflammation through supplementation with specific nutrients, such as arginine, omega-3 fatty acids, nucleotides, and glutamine. Arginine is a precursor to nitric oxide, which improves blood flow and supports wound healing [101,102]. Recent studies confirm that preoperative arginine supplementation can reduce the risk of infectious complications [103]. Omega-3 fatty acids have anti-inflammatory effects, which are particularly important for patients with chronic conditions such as cancer or heart disease. Recent meta-analyses suggest that preoperative omega-3 supplementation can reduce inflammation and improve preoperative outcomes such as C-reactive protein, interleukin-6, and white blood cell counts [104,105]. In a study by Ryan et al., omega-3 supplementation in lung cancer patients reduced inflammation, improved appetite, and slowed muscle loss. Patients receiving omega-3 also had better quality of life and lower risk of postoperative complications [106]. Studies showed that 18 g of glutamine supplementation 7 days before surgery can reduce inflammation, with lower levels of C-reactive protein and interleukin-6 observed in laboratory tests [107]. Increasing evidence suggests that combining immunomodulatory supplements with probiotics can further enhance immune function [108]. Patients receiving immunomodulatory supplements had shorter hospital stays and better biochemical parameters, including higher albumin and prealbumin levels [13].

### 4.3. The Role of ERAS in Optimizing Nutritional Support

The Enhanced Recovery After Surgery (ERAS) program is a multidisciplinary strategy aimed at accelerating postoperative recovery by optimizing pre-, intra-, and postoperative care [109]. In terms of nutrition, ERAS emphasizes early nutritional preparation—identifying patients at risk of malnutrition and implementing nutritional interventions 5–7 days before surgery. Immunomodulation is particularly recommended for patients with cancer or chronic conditions [101,102]. Optimizing nutritional status by ensuring adequate protein and calorie intake is crucial for improving nutritional and immune function. Recent studies suggest that preoperative nutritional optimization can reduce complication risks by up to 40%, and early postoperative nutrition (encouraging oral intake within the first day after surgery) accelerates intestinal recovery and reduces risks such as intestinal obstruction or infections [110]. Regular monitoring of nutritional status (e.g., albumin and prealbumin levels) is key for adjusting nutritional interventions. Modern methods such as bioelectrical impedance analysis (BIA) allow for more precise nutritional assessments [13].

A study by Huang et al. demonstrated that preoperative nutritional preparation using the ERAS protocol for lung tumor resection improved nutritional status and reduced surgical complications such as lung infections and pleural effusion. Patients in the intervention group showed better nutritional status, with higher prealbumin, albumin, and hemoglobin levels compared to the control group [13]. Another study showed that implementing the ERAS protocol, including early postoperative nutrition and preoperative nutritional optimization, reduced complication risks by 40% and shortened hospital stays by an average of 2–3 days. ERAS patients also had better functional outcomes and faster return to daily activities [109,110].

## 5. Practical Guidance for Pulmonologist, Oncologists, and Thoracic Surgeons


Nutritional and Sarcopenia Assessment
Mandatory nutritional screening and standardized diagnosis: This includes performing systematic nutritional risk screening at surgical qualification using validated tools (e.g., NRS 2002 and MUST). In patients identified as at-risk, the Global Leadership Initiative on Malnutrition (GLIM) criteria should be applied to confirm the diagnosis of malnutrition, requiring at least one phenotypic and one etiologic criterion;Comprehensive nutritional assessment: this includes conducting detailed evaluations, including dietary interview, physical examination (BMI, calf and arm circumference, handgrip strength), laboratory investigations (albumin, prealbumin, CRP), and body composition analysis (BIA, DXA, CT, or MRI when available);Early detection of sarcopenia: this includes assessing skeletal muscle mass and strength systematically, utilizing validated thresholds (e.g., SMI values on CT or DXA and handgrip strength measures) to diagnose sarcopenia in its early stages;Continuous perioperative monitoring: this includes reassesses nutritional status and muscle mass throughout the perioperative period;Multidisciplinary team approach: this includes integrating dietitians, physiotherapists, and rehabilitation specialists into the oncological care team to optimize nutritional support and enhance functional recovery.
High-Protein Diet
Individualize protein requirements: age, body weight, health status, and physical activity levels should be considered when determining daily protein needs. For most patients, 1.2–1.5 g/kg body weight/day is optimal, but elderly or severely malnourished patients may require 1.5–2.0 g/kg body weight/day;Diverse protein sources: a variety of protein sources should be recommended, including animal-based (poultry, fish, eggs, and dairy) and plant-based (legumes, tofu, and quinoa).
Oral Nutritional Supplements
Early intervention: oral nutritional supplements should be implemented early for patients with appetite loss, swallowing difficulties, or other issues preventing adequate food intake;Combine with physical activity: if possible, combining protein supplementation with resistance exercises should be recommended to improve muscle mass and strength before surgery;Monitor outcomes: body weight and composition should be regularly assessed to evaluate the effectiveness of supplementation and adjust dosages as needed.Enteral Nutrition
Choose the right access method: Nasogastric tubes or PEG should be considered for patients with swallowing difficulties or chronic malnutrition. PEG is preferred for long-term nutritional support;Early implementation: enteral nutrition should be initiated 7–10 days before surgery in malnourished patients to reduce infection risks and improve wound healing;Monitor for complications: patients should be watched for risks such as diarrhea, tube obstruction, or aspiration pneumonia, and they should be monitored regularly.
Parenteral Nutrition
Limited use: parenteral nutrition should be used only when enteral nutrition is not feasible, such as in cases of severe malabsorption, intestinal obstruction, or short bowel syndrome;Intensive monitoring: due to risks such as sepsis, metabolic disturbances, and liver damage, parenteral nutrition requires regular laboratory monitoring and tailored therapy.
Perioperative Immunomodulation
Arginine supplementation: arginine supplementation should be considered before surgery to improve blood flow and wound healing, reducing the risk of infectious complications;Omega-3 fatty acids: omega-3 supplementation should be recommended for patients with chronic conditions (e.g., cancer) to reduce inflammation and improve preoperative outcomes;Glutamine: glutamine supplementation should be considered for patients at high risk of inflammation, as it can reduce markers such as C-reactive protein and interleukin-6;Combine with probiotics: increasing evidence suggests that combining immunomodulatory supplements with probiotics can enhance immune function and shorten hospital stays.



## 6. Summary

Assessing the nutritional status of lung cancer patients is crucial for improving surgical outcomes. Various assessment methods, such as nutritional history, physical examination, laboratory tests, malnutrition risk scales, and body composition analysis, allow for a comprehensive evaluation of the patient’s condition. Due to the limitations of individual methods, a multifaceted approach combining different diagnostic tools is recommended.

Malnutrition significantly increases the risk of postoperative complications, including lung infections, respiratory failure, and prolonged hospitalization. Malnourished patients have poorer prognoses in terms of overall survival and disease-free survival. Muscle loss weakens patients’ ability to undergo rehabilitation, reducing their quality of life after surgery.

The clinical implications of these findings highlight the need for standardized nutritional protocols in the care of lung cancer surgery patients. Mandatory nutritional assessments at the time of surgical qualification can identify at-risk patients early and allow for appropriate interventions. The inclusion of the ERAS protocol, which emphasizes preoperative nutritional optimization, reduced fasting periods, and postoperative metabolic support, is also recommended.
Areas for Future Research:
Lack of standardized nutritional assessment methods: despite the availability of various tools, there is a lack of unified guidelines for their use across different disease stages and patient groups;Limitations in body composition assessment: methods such as bioelectrical impedance analysis (BIA) and dual-energy X-ray absorptiometry (DXA) are unavailable in many centers, making precise assessment of muscle and fat mass difficult;Insufficient understanding of cancer cachexia: the mechanisms driving cachexia and its impact on treatment outcomes require further research, particularly in the context of nutritional and pharmacological interventions;Lack of long-term studies on nutritional interventions: most available data focus on short-term effects, while the long-term benefits of nutritional optimization remain unclear;Limitations in immunomodulation use: despite promising results, there are no clear guidelines on dosing and duration for immunomodulatory supplements (e.g., arginine and omega-3 fatty acids).


## 7. Conclusions

The findings underscore the need for standardized nutritional protocols in the care of lung cancer surgery patients. Mandatory nutritional assessments at the time of surgical qualification can identify at-risk patients early and allow for appropriate interventions. The inclusion of the ERAS protocol, which emphasizes preoperative nutritional optimization, reduced fasting periods, and postoperative metabolic support, is also recommended. Implementing comprehensive nutritional strategies will not only improve patient outcomes but also reduce complication rates and hospital stays, leading to lower treatment costs and increased healthcare system efficiency.

## Figures and Tables

**Figure 1 jcm-14-03197-f001:**
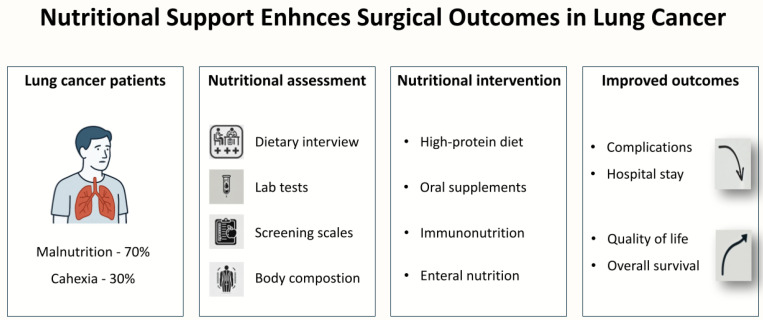
Nutritional Support Enhances Surgical Outcomes in Lung Cancer.

**Table 1 jcm-14-03197-t001:** Advantages and Disadvantages of Nutritional Assessment Methods.

Assessment Method	Description	Advantages	Disadvantages
Dietary Interview	Detailed conversation with the patient about dietary habits, preferences, eating problems, and changes in body weight.	Easy to conduct, inexpensive, and allows for initial identification of patients at risk of malnutrition.	Subjective; depends on the patient’s memory and ability to report accurately.
Physical Examination	Measurement of body weight, height, calf and arm circumference, assessment of muscle strength, and clinical signs of malnutrition.	Quick and allows for detection of malnutrition and cancer cachexia.	Effectiveness depends on the examiner’s experience; may be subjective.
Laboratory Tests	Assessment of albumin, prealbumin, transferrin, peripheral lymphocyte count, cholesterol, and C – reactive Protein-CRP levels.	Objective data on nutritional status.	Results may be influenced by inflammatory conditions and comorbidities.
Scales and Questionnaires	Mini Nutritional Assessment-MNA, Subjective Global Assessment-SGA, Patient-Generated Subjective Global Assessment-PG-SGA, Nutritional Risk Screening 2002-NRS 2002, and Malnutrition Universal Screening Tool-MUST.	Quick, non-invasive, and easy to use.	May not consider all aspects of nutritional status.
Body Composition Analysis	Bioelectrical impedance analysis-BIA, Dual-energy X-ray absorptiometry-DXA, computed tomography -CT, and magnetic resonance imaging -MRI.	Precise data on muscle and fat mass.	Requires specialized equipment and expertise.
Controlling Nutritional Status-CONUT and C-reactive protein-albumin-lymphocyte-CALLY Index	Assessment of albumin, peripheral lymphocyte count, cholesterol, and CRP levels.	Simple, based on laboratory parameters, and useful for prognosis assessment.	Do not consider anthropometric parameters.

## Data Availability

Not applicable.

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
