# Peer review of "The Impact of Nutritional Support on Outcomes of Lung Cancer Surgery—Narrative Review"

_jcm, 2025, doi:10.3390/jcm14093197_

Round 1

Reviewer 1 Report

Comments and Suggestions for Authors

This is a narrative review that examines the impact of nutritional status on lung cancer surgery and explores the role of nutritional assessment and intervention strategies for optimizing surgical outcomes.

The review contributes with a general view of this topic, but does not offer a deep insight.

The paper talks about malnutrition but it should cover sarcopenia too.

In line 72 authors say that malnutrition is defined as a loss of 10% of body weight or a low body mass index (BMI<18). Although they add that this definition may slightly differ depending on the guidelines (e.g., ASPEN, ESPEN), it should be useful to readers to deep in the definitions of malnutrition, because the definition given by authors is uncomplete and inaccurate. GLIM criteria should be included in this explanation of malnutrition.

In line 127 authors say that laboratory tests provide objective data on nutritional status. This statement has to be adequately discussed because of the limitations of laboratory tests in determining the nutritional status.

In line 142 authors introduce the scales and questionnaires. GLIM criteria should be added.

In line 159 authors describe the methods of body composition analysis. The explanation is very poor, this should be extended. For example, in BIA, the authors should describe the utility of phase angle and the detection of low muscle mass.

In line 343 the authors offer a series of Practical Tips for Pulmonologist, Oncologists and Thoracic Surgeons. They should add tips about the diagnosis of malnutrition and/or sarcopenia, not only about treatment features.

The summary and conclusions are coherent with the content of the manuscript.

The references are recent in general.

Author Response

Dear Reviewer,

We greatly appreciate your positive feedback on our manuscript. Thank you very much for all your valuable comments and suggestions. We have addressed all your points carefully and modified the manuscript accordingly. We hope that the revisions have improved the quality and depth of our work. Please find our point-by-point responses below.

Comment 1: The review contributes with a general view of this topic but does not offer a deep insight. The paper talks about malnutrition but it should cover sarcopenia too.

Response and Changes 1: Thank you for your comment. We agree that sarcopenia is an important aspect when discussing nutritional status in lung cancer patients. Therefore, we have significantly expanded the manuscript by including a detailed discussion on sarcopenia, its assessment methods (e.g., handgrip strength, BIA, CT-based skeletal muscle index), and its clinical implications for surgical outcomes. The practical tips for clinicians have also been updated to include systematic screening and early diagnosis of sarcopenia.

Comment 2: In line 72, authors define malnutrition as a loss of 10% of body weight or BMI<18. Although they mention ASPEN/ESPEN, a deeper explanation is needed. GLIM criteria should be included because the current definition is incomplete and inaccurate.

Response and Changes 2: Thank you for this suggestion. We fully agree. We have revised the section on the definition of malnutrition to include a detailed description of the Global Leadership Initiative on Malnutrition (GLIM) criteria, which require both phenotypic and etiologic indicators. We have also discussed how GLIM complements traditional definitions and enhances the precision of malnutrition diagnosis, especially in oncology patients.

Comment 3: In line 127, authors say that laboratory tests provide objective data on nutritional status. This needs to be discussed more critically because of the limitations of lab tests.

Response and Changes 3: Thank you for pointing this out. We have expanded the discussion on laboratory tests, explicitly highlighting their limitations. We explain that lab parameters (such as albumin, prealbumin, transferrin, lymphocytes) can be influenced by inflammation and other comorbidities, and therefore, they should always be interpreted together with clinical assessment and screening tools to avoid misclassification.

Comment 4: In line 142, authors introduce scales and questionnaires. GLIM criteria should be added.

Response and Changes 4: Thank you for the comment. We have revised this section and added a detailed explanation of the GLIM criteria, including the two-step diagnostic approach (screening + phenotypic/etiologic assessment) and its validated role in oncology populations. We also discussed its advantages over classical questionnaires.

Comment 5: In line 159, body composition analysis is described, but very briefly. It should be expanded, e.g., description of BIA phase angle and low muscle mass detection.

Response and Changes 5: Thank you for this important suggestion. We have significantly expanded the section on body composition methods. We described BIA in more detail, including the importance of phase angle as a prognostic marker, detection thresholds for muscle mass, and its clinical relevance. We also added detailed explanations of DXA, CT, and MRI in body composition analysis, with specific oncologic thresholds.

Comment 6: In line 343, the authors offer Practical Tips but focus mainly on treatment. They should add tips about diagnosing malnutrition and sarcopenia.

Response and Changes 6: Thank you for the valuable feedback. We have expanded the Practical Tips section to emphasize the importance of systematic nutritional and sarcopenia screening, including the use of validated tools (NRS 2002, MUST, GLIM criteria), assessment of handgrip strength, and body composition analysis (BIA/CT). We also highlighted the importance of integrating dietitians and rehabilitation specialists into the perioperative team.

We hope that these revisions have addressed all the concerns raised and have significantly improved the quality and depth of our manuscript.

Reviewer 2 Report

Comments and Suggestions for Authors

The role of nutrition for the enhanced recovery after surgery (ERAS) is reviewed based on an extensive literature search. The proposed protocols interrogating the nutritional status of the lung cancer patients combined with objective medical analyses followed by correction of nutrient deficit is of interest. This may enhance recovery and long-term prognosis after lung cancer surgery.

This is a great and important translational review.

Suggestion:

A graphical abstract with the main issues on the nutritional status of cancer patients is recommended to  alert readers and the busy surgeons.

Author Response

Dear Reviewer, we greatly appreciate your positive and encouraging feedback on our manuscript. Thank you very much for recognizing the importance and translational value of our review. We are grateful for your constructive suggestion and have addressed it as detailed below.

Comment: a graphical abstract with the main issues on the nutritional status of cancer patients is recommended to alert readers and the busy surgeons.

Response and Changes: Thank you for this excellent suggestion. We fully agree that a graphical abstract summarizing the key aspects of nutritional status and interventions could enhance the accessibility of our review, particularly for busy clinicians.

Accordingly, we have prepared a graphical abstract highlighting the following key points:

  • Prevalence and consequences of malnutrition and sarcopenia in lung cancer patients
  • Essential tools for nutritional assessment (e.g., screening, clinical evaluation, body composition analysis)
  • Impact of malnutrition on surgical outcomes
  • Overview of nutritional support strategies (high-protein diet, oral supplements, enteral and parenteral nutrition, perioperative immunomodulation)
  • The role of nutrition within the Enhanced Recovery After Surgery (ERAS) protocol

The graphical abstract has been included in the revised version of the manuscript as requested.

We sincerely appreciate your valuable input, which helped us to improve the manuscript further.

Round 2

Reviewer 1 Report

Comments and Suggestions for Authors

Congratulations to the authors. They have considered all the comments of the previous review and they have implemented them. The current work is suitable for publication.